# Transcriptomic and Proteomic Analyses of *Myzus persicae* Carrying Brassica Yellows Virus

**DOI:** 10.3390/biology12070908

**Published:** 2023-06-25

**Authors:** Meng-Jun He, Deng-Pan Zuo, Zong-Ying Zhang, Ying Wang, Cheng-Gui Han

**Affiliations:** Ministry of Agriculture and Rural Affairs Key Laboratory of Pest Monitoring and Green Management, College of Plant Protection, China Agricultural University, Beijing 100193, China; s20193192583@cau.edu.cn (M.-J.H.); b20173190806@cau.edu.cn (D.-P.Z.); zhangzongying@cau.edu.cn (Z.-Y.Z.); yingwang@cau.edu.cn (Y.W.)

**Keywords:** Brassica yellows virus (BrYV), *Myzus persicae*, transcriptome, proteome

## Abstract

**Simple Summary:**

Brassica yellows virus (BrYV) transmitted by the green peach aphid belongs to the genus *Polerovirus* and mostly damages the crucifer crops in East Asia. To reveal the networks of aphid gene response to BrYV stress, comparative transcriptome and proteome approaches were performed to identify significantly putative regulators involved in BrYV stress. Based on the results of RNA Sequencing (RNA-Seq) coupled with quantitative proteomic analysis on BrYV-carrying and BrYV-free aphids, 1266 differently expressed genes (980 upregulated and 286 downregulated DEGs) and 18 differently expressed proteins (12 upregulated and 6 downregulated DEPs) were identified in BrYV-carrying aphids. Enrichment analysis indicated that these DEGs and DEPs were primarily involved in epidermal protein synthesis, phosphorylation, and various metabolic processes. Interestingly, the expressions of a number of cuticle proteins and tubulins were substantially upregulated in viruliferous aphids. To sum up, these findings provide a crucial clue for screening key vector factors involved in the process of virus circulation in aphids and exploring the molecular mechanism of transmission of BrYV by the green peach aphid.

**Abstract:**

Viruses in the genus *Polerovirus* infect a wide range of crop plants and cause severe economic crop losses. BrYV belongs to the genus *Polerovirus* and is transmitted by *Myzus persicae*. However, the changes in transcriptome and proteome profiles of *M. persicae* during viral infection are unclear. Here, RNA-Seq and TMT-based quantitative proteomic analysis were performed to compare the differences between viruliferous and nonviruliferous aphids. In total, 1266 DEGs were identified at the level of transcription with 980 DEGs being upregulated and 286 downregulated in viruliferous aphids. At the protein level, among the 18 DEPs identified, the number of upregulated proteins in viruliferous aphids was twice that of the downregulated DEPs. Enrichment analysis indicated that these DEGs and DEPs were mainly involved in epidermal protein synthesis, phosphorylation, and various metabolic processes. Interestingly, the expressions of a number of cuticle proteins and tubulins were upregulated in viruliferous aphids. Taken together, our study revealed the complex regulatory network between BrYV and its vector *M. persicae* from the perspective of omics. These findings should be of great benefit to screening key factors involved in the process of virus circulation in aphids and provide new insights for BrYV prevention via vector control in the field.

## 1. Introduction

Numerous plant viruses are transmitted by specific insect vectors, most of which belong to the order Hemiptera. These insects have piercing–sucking mouthparts and their feeding organs can access the plant cell cytoplasm by breaking through the cell wall to achieve efficient transmission of the virus to new hosts [1,2,3]. Aphids are considered to be the most frequent and effective insect vectors for virus transmission because they can transmit over 50% of arthropod-borne viruses, which cause tremendous economic impact around the world.

Poleroviruses (family *Solemoviridae*) are a group of single-stranded positive-sense RNA viruses that cause yellowing or leafroll symptoms, leading to significant economic crop losses. In general, these viruses are restricted to the phloem cells of their hosts and transmitted by aphids in a circulative manner. Furthermore, a high level of vector-specificity exists between poleroviruses and aphids [4,5]. Interestingly, a new whitefly (*Bemisia tabaci*)-transmitted polerovirus infecting bell peppers has been reported recently, which caused heavy economic losses for peppers cultivated in Israel [6], highlighting that aphids are not the only vectors of poleroviruses. Previous work has described the circulation pathway of the poleroviruses in aphids. Ingestion of viruses begins when aphids feed on virus-infected host plants using their stylets, at which point the virions move upward along the aphid alimentary canal, arriving first at the gut epithelial cells via endocytosis in clathrin-coated vesicles, followed by transport in the vesicles from the apical to the basal pole of the cell [7,8]. After reaching the basal pole, virions could be released by fusion of the vesicles with the basal lamina, then migrate freely into the hemolymph [5,9]. Subsequently, the virions could cross the basal lamina surrounding the accessory salivary gland cells and are released into the salivary duct where they are transported by coated vesicles or tubular vesicles. Finally, they are introduced into new plant hosts along with the secretion of saliva during feeding [3,5,10], at which point a new circulation of virus infection and spread of disease occur as a result of virus transmission.

Brassica yellows virus (BrYV) is a tentative newly identified species in the genus *Polerovirus* and it resembles turnip yellows virus (TuYV). Three distinct genotypes of BrYV (A, B and C) had been identified in China, with Br-C being considered the most dominant genotype [11,12]. When a 35S promoter-derived expression cassette containing the full-length cDNA of Br-C was successfully transformed into the *Arabidopsis thaliana* Col-0 ecotype, two lines, 111 and 412, of these transgenic *Arabidopsis* plants were generated for further study [13]. *M. persicae* has been reported as the vector of BrYV and a simple novel method was established for the acquisition and transmission of BrYV from transgenic line 412 *Arabidopsis* plants or frozen infected leaves [14]. Polerovirus particles are composed of the major coat protein (CP) and a few copies of a minor readthrough protein (RTP) which is a fusion protein with the CP at its N-terminus and the readthrough domain (RTD) at its C-terminus. It has been reported that mutations in the CP or RTP of potato leaf roll virus (PLRV) can seriously affect the transmission efficiency of aphids, virus movement, and virus accumulation [15,16]. For circulative phytoviruses, when they move through the insect vector, the interaction between the virus and vector proteins would have occurred to overcome gut and salivary gland barriers to achieve successful transmission. For example, membrane alanyl aminopeptidase N (APN) was confirmed as the first receptor for pea enation mosaic virus (PEMV) coat protein (CP) in the gut of the pea aphid (*Acyrthosiphon pisum*). The CP could bind to APN, competing with GBP3.1, a peptide previously reported to prevent PEMV from entering the haemocoel, to promote virus transport in aphids [17,18]. In addition to APN, several aphid proteins were also found to have the ability to bind to purified virions in vitro, such as SaM35 (*Sitobion avenae* protein binding isolate MAV, MW 35 kDa) and SaM50 (*S. avenae* protein binding isolate MAV, MW 50 kDa) proteins isolated from the head tissue of *S. avenae*, which showed a high affinity for barley yellow dwarf virus (BYDV-MAV) [19]. The receptor for activated C kinase 1 (Rack-1), glyceraldehyde-3-phosphate dehydrogenase (GAPDH3), and actin of *M. persicae* could bind to purified TuYV particles in vitro [20]. Similar to previous studies, luciferase and cyclophilin proteins of greenbug *Schizaphis graminum* were found to be specifically associated with cereal yellow dwarf virus-RPV (CYDV-RPV), where the authors associated these two proteins only with endocytosis, with the real receptors of persistent-circulative viruses in aphids needing to be further studied [21]. To date, the viral determinants and their corresponding receptors in aphids, that regulate the specific interaction between BrYV and *M. persicae,* have not yet been determined.

Deep-sequencing techniques have provided new insights into complex insect–virus interactions and can identify genes or proteins of vectors potentially involved in virus acquisition and transmission. Comparative proteomic analysis of Mediterranean (MED) whitefly infected by tomato yellow leaf curl virus (TYLCV) or papaya leaf curl China virus (PaLCuCNV) showed that 20S proteasome subunits involved in virus degradation were upregulated only in PaLCuCNV-infected whiteflies, which may explain why whiteflies transmitted TYLCV more efficiently than PaLCuCNV [22]. Using iTRAQ labeling analysis, the nucleolar RNA-binding protein esf2 and the protein associated with mRNA decay (ZFP36L1) were found to be differentially expressed in rice stripe virus (RSV)-infected small brown planthopper (SBPH), *Laodelphax striatellus*, compared with RSV-free *L. striatellus*, which played critical roles in insect physiology and RSV burden, respectively [23]. For aphids, using isobaric tags for relative and absolute quantitation (iTRAQ) and yeast two-hybrid (YTH) methods, many proteins were identified as potentially being involved in the transmission of barley yellow dwarf virus-GPV by the bird cherry–oat aphid, *Rhopalosiphum padi*, such as the proteasome, cuticle proteins, vesicle-associated membrane protein (VAMP), synaphin, and the cytoskeleton [24]. Combined transcriptomic and proteomic analysis of *M. persicae* infected with cucumber mosaic virus (CMV) have identified some vector regulators associated with virus transmission, such as cuticle proteins, ribosomal proteins, and cytochrome P450 enzymes [25]. More than that, previous studies have shown that the titer of the primary endosymbiont *Buchnera aphidicola* was reduced and 134 differentially expressed genes were detected in PLRV-carrying aphids compared with virus-free aphids [26]. Thus, to better understand the effect of BrYV on the green peach aphid, we used two approaches, RNA-Seq and Tandem Mass Tag (TMT)-based quantitative proteomic analysis, to compare the differences between BrYV-carrying and BrYV-free aphids at the mRNA and protein levels. Our findings will provide new insights into the interactions between BrYV and its vector, *M. persicae*. Combined enrichment analysis of Gene Ontology (GO) terms and Kyoto Encyclopedia of Genes and Genomes (KEGG) pathways of DEGs and DEPs also offer a tremendous opportunity to identify genes which may be involved in virus transmission, for further functional validation.

## 2. Materials and Methods

### 2.1. Aphid Culture and Plant Materials

Apterous *M. persicae* individuals were derived from a single clonal lineage reared on turnip variety ‘Yamei No. 1’ in cages. The 3- to 4-week-old seedlings of *A. thaliana* line 412, harboring the BrYV-C full-length cDNA clone, were grown in a greenhouse and were used as viral inoculum as previously described [14]. In addition, the full-length infectious BrYV-A clone was transferred to *Agrobacterium tumefaciens* C58C1 and inoculated into *A. thaliana* for two weeks, after which the systemic leaves were detected using reverse transcriptase PCR (RT-PCR). These two materials were used as virus-inoculation materials for aphid feeding. All plants were grown in a greenhouse at 22–24 °C with a photoperiod cycle of 14 h light/10 h darkness and 60% relative humidity.

### 2.2. Preparation of Samples

To obtain viruliferous and nonviruliferous aphids for RNA-Seq analysis, approximately 400 s to fourth instar individuals of apterous aphids were collected for transfer to the transgenic line 412 plants or healthy *A. thaliana* plants for feeding for 3 days; then, the viruliferous and nonviruliferous aphids, respectively, were incubated on healthy *A. thaliana* plants for 2 weeks. Six samples were collected, one for each of the three biological replicates, each consisting of fifty individuals. After RNA extraction, a portion of vital RNA was confirmed using RT-PCR (Appendix A). Subsequently, the remaining samples were stored at −80 °C for RNA sequencing.

To prepare viruliferous and nonviruliferous aphids for TMT analysis, the full-length infectious BrYV-A clone inoculated into *A. thaliana* plants was used as a source material for aphid feeding to acquire BrYV. After 7 days, approximately 400 s to fourth instar individuals of apterous aphids were collected for protein extraction; control aphids fed on healthy *A. thaliana* plants. Three biological replicates were performed, and some aphids were randomly selected and analyzed using RT-PCR (Appendix A); after the detection, six samples were stored at −80 °C for TMT analysis.

### 2.3. RNA Extraction and cDNA Library Construction

Total RNA was extracted using TRIzol reagent (Invitrogen, San Diego, CA, USA) and RNA integrity was assessed using the RNA Nano 6000 Assay Kit of the Bioanalyzer 2100 system (Agilent Technologies, Palo Alto, CA, USA). Sequencing libraries were generated using the NEBNext^®^ UltraTM RNA Library Prep Kit for Illumina (NEB, Ipswich, MA, USA) in accordance with the instructions of the manufacturer. Briefly, mRNA was purified from total RNA using poly-T oligo-attached magnetic beads. Subsequently, fragmentation was carried out using divalent cations and the mRNA fragments were used as the template to synthesize the two strands of cDNA. In order to select cDNA fragments of preferentially 370–420 bp in length, the library fragments were purified using an AMPure XP system (Beckman Coulter, Beverly, CA, USA). The purified and adaptor-ligated cDNA was subjected to PCR amplification. Finally, PCR products were purified using the AMPure XP system and quality was assessed on the Agilent Bioanalyzer 2100 system.

### 2.4. Transcriptomic Analysis

Clean reads were obtained by removing reads containing adapter sequences, uncertain base, and low-quality reads from the raw data. At the same time, the Q20, Q30, and GC content of the clean data were calculated. All downstream analyses were based on the high-quality clean data. Differentially expressed genes were analyzed using the ‘DESeq2’ package in R (1.20.0) [27]. The resulting *p*-values were adjusted using the Benjamini and Hochberg’s approach to control the false discovery rate. |log_2_ (fold change, FC)| > 1 and *p* adj < 0.05 were used as the criteria to identify significantly upregulated and downregulated genes.

The GO “http://geneontology.org/ (accessed on 15 April 2023)”annotation contains the aspects’ biological processes, cell components, and molecular functions. KEGG “http://www.genome.jp/kegg/ (accessed on 15 April 2023)” is a database resource for understanding the high-level functions and utilities of the biological system. Statistical enrichment analysis was used by the ‘clusterProfiler’ package in R to identify the GO terms and KEGG pathways. A corrected *p*-value < 0.05 was considered to be significantly enriched by differentially expressed genes.

### 2.5. Protein Extraction, TMT Labeling, and LC-MS/MS

Buffer composed of 4% SDS, 100 mM Tris-HCl, and 1 mM dithiothreitol (DTT), (pH7.6) was used for sample lysis and protein extraction. An aliquot (200 μg) of proteins from each sample was digested using trypsin (Promega, Madison, WI, USA). After SDS-PAGE, 100 μg peptide mixture of each sample was labeled using TMT reagent (Thermo Fisher Scientific, Waltham, MA, USA), according to the instructions of the manufacturer. The TMT tags 129,130, and 131 were used to label BrYV-carrying aphids; the control aphids were labeled as 126,127, and 128. Then, labeled peptides were fractionated using the High pH Reversed-Phase Peptide Fractionation Kit (Thermo Fisher Scientific, Waltham, MA, USA). Subsequently, liquid chromatography–tandem mass spectrometry (LC-MS/MS) analysis was performed on a Q Exactive mass spectrometer (Thermo Fisher Scientific, Waltham, MA, USA) that was coupled to Easy nLC (Thermo Fisher Scientific, Waltham, MA, USA). Finally, the MS raw reads for each sample were searched using the MASCOT engine “matrixscience.com (accessed on 25 July 2021)” embedded into Proteome Discoverer 1.4 software for identification and quantitation analysis.

### 2.6. Proteomic Analysis

|fold change, (FC)| > 1 and *p*-value < 0.05 were used as the criteria to identify differentially expressed proteins between treatment and control aphids. The proteins were filtered using an FDR (false discovery rate) ≤0.01. The subcellular localization of DEPs was analyzed using CELLO “http://cello.life.nctu.edu.tw/ (accessed on 25 July 2021)”.

GO “http://geneontology.org/ (accessed on 15 April 2023)” terms that had been mapped and sequences were annotated using the software program Blast2GO. Subsequently, the proteins studied were blasted against the online KEGG database “http://geneontology.org/ (accessed on 15 April 2023)” to further investigate these proteins by assigning them to pathways. Only functional categories and pathways with *p*-values under the threshold of 0.05 were considered to be significant.

### 2.7. RT-qPCR Verification

Ten candidate genes of *M. persicae* were selected for validation of RNA-*S*eq data and seven DEGs involved in some significant pathways were selected, including “phagosome” (tubulin alpha-2 chain-like), carbon metabolism (probable isocitrate dehydrogenase [NAD] subunit alpha and trifunctional enzyme subunit alpha), oxidative phosphorylation (cytochrome c oxidase subunit 5B), Glycolysis/Gluconeogenesis (aldose 1-epimerase-like), Galactose metabolism (beta-galactosidase-like), and starch and sucrose metabolism (uncharacterized LOC111032711) pathways. In addition, three differentially expressed genes were also verified using RT-qPCR, including cuticle protein 65-like, general odorant-binding protein 28a, and cuticle protein 21-like, which may play an important role in virus circulation and transmission.

Total RNA was extracted using TRIzol reagent (Invitrogen, San Diego, CA, USA) and RT-PCR was performed as described previously by Zuo [14]. Reverse transcription quantitative PCR (RT-qPCR) analysis was performed to confirm the relative expression of DEGs and DEPs, using GoTaq^®^ qPCR Master Mix (Promega, Madison, WI, USA). Each reaction system contained 10 µL 2 × SYBR Green real-time PCR Master Mix, 0.5 μL forward primer (10 μM), 0.5 μL reverse primer (10 μM), 1 μL cDNA, and 8 μL RNase-free ddH2O. The reaction program was as follows: 95 °C for 2 min, followed by 40 cycles of 95 °C for 15 s, 60 °C for 1 min, and 72 °C for 30 s. The actin gene (ACCESSION: XM_022309797) was used as an internal reference for relative gene expression analysis. The experiment was repeated at least three times. Primers used in the qPCR validation are shown in Appendix A.

## 3. Results

### 3.1. Basic Quantitative Statistics of Viruliferous and Nonviruliferous Aphids

To explore the differences in gene expression levels between BrYV-carrying and BrYV-free aphids, we performed RNA-Seq and TMT-based quantitative proteomic analysis at both the mRNA and protein levels. The cDNA libraries of viruliferous and nonviruliferous aphids were sequenced, generating 53,943,971 and 56,380,878 raw reads, respectively. After removal of the low-quality reads, a total of 53,190,385 and 55,432,735 clean reads were obtained, respectively, within which the Q30 values were calculated as 92.82% and 93.36% (Appendix A), indicating the high quality of the sequencing data. Subsequently, the clean reads were compared with the reference genome of the green peach aphid on NCBI “https://www.ncbi.nlm.nih.gov/genome/?term=myzus%20persicae%20 (accessed on 18 April 2023)”, where 91.79% and 92.12% reads, respectively, were found to be unique mapped reads (Table 1).

The total number of mass spectra detected in *M. persicae* using quantitative proteomic analysis was 725,126 under the filter criterion of the false discovery rate (FDR) ≤ 0.01. Of the spectra identified, we found that the number of matched spectra was 71,746. In total, 40,482 peptides were identified, and the number of unique peptides was 34,831, whereas the number of proteins identified was 5660. Among these identified proteins, 5567 were quantified proteins, which indicated that more than half of the biological replicates had the strength value of this protein (Table 2).

### 3.2. Analysis of Gene Expression Differences of M. persicae in Response to BrYV Stress

For transcriptomic difference analysis, using |log_2_ (fold change, FC)| > 1 and *P p* adj < 0.05 as the criteria, a total of 1266 differentially expressed genes were identified between the viruliferous and nonviruliferous aphids, with 980 upregulated and 286 downregulated genes in viruliferous aphids (Figure 1a,b). In addition, the extent of log_2_ FC ranged from −3.3 to 15.3, and more than 97% of the downregulated DEGs ranged from −2 to −1, indicating that either the number or the fold change of upregulated DEGs was significantly higher than that of the downregulated DEGs. It is worth noting that at least 25 cuticle proteins that may be related to virus transmission were all upregulated in response to BrYV stress.

For proteomic difference analysis, we used 1.2-fold change (upregulated > 1.2 or downregulated < 0.83) and *p* value < 0.05 as the criteria by which to select differentially expressed proteins. For *M. persicae* fed on BrYV-infected line 412 transgenic plants, 18 DEPs were identified, with 12 upregulated and 6 downregulated (Figure 1c,d). Indeed, over the two omics, the majority of the DEGs and DEPs were upregulated in the presence of BrYV.

### 3.3. Enrichment Terms and Pathway Analysis of M. persicae in Response to BrYV Stress

To better understand the functions of DEGs and the pathways involved in exposure of *M. persicae* to BrYV, enrichment analyses using the GO and KEGG databases were performed using *p* < 0.05 as the criteria. A total of 1266 DEGs were categorized into 33 GO terms under the aspects biological process (BP), molecular function (MF), and cellular component (CC). Specifically, terms “oxidoreductase activity”, “structural constituent of cuticle”, “structural molecule activity”, and “peptidase activity” were highly enriched in 342 DEGs in aspect MF. More interestingly, the only upregulated DEGs were enriched with respect to the structural constituent of cuticle and structural molecule activity categories, which may be important for *M. persicae* in response to BrYV stress. A total of 83 DEGs (66 upregulated and 17 downregulated) and 56 DEGs (54 upregulated and 2 downregulated) were enriched in the BP and CC aspects, respectively. “Proteolysis” was the most common term in the BP aspect and “cytoskeletal part”, “cytoskeleton”, and “catalytic complex” were the major terms enriched under the CC aspect (Figure 2a).

To investigate which biological pathways were active in viruliferous *M. persicae*, the DEGs were mapped to eighty-six pathways in KEGG, of which seven pathways were significantly enriched, including “Carbon metabolism” (thirteen DEGs), “Oxidative phosphorylation” (twelve DEGs), “Phagosome” (eleven DEGs), “Biosynthesis of amino acids” (nine DEGs), “Glycolysis/Gluconeogenesis” (seven DEGs), “Galactose metabolism” (five DEGs), and “Starch and sucrose metabolism pathways” (five DEGs). Among these pathways, “Oxidative phosphorylation”, “Glycolysis/Gluconeogenesis”, and “Galactose metabolism” were enriched with respect to only upregulated genes (Table 3).

To reveal the functions and pathways of identified and quantified DEPs, subcellular localization analysis was performed using CELLO software. The results showed that the majority of DEPs were localized in the nucleus and at the plasma membrane (Figure 3). In the GO analysis, 127 DEPs were annotated according to biological processes, 54 to molecular functions, and 38 to cellular component aspects. Compared with nonviruliferous aphids, “phosphorylation”, “phosphorus metabolic process”, “phosphate-containing compound metabolic process”, and “metabolic process” were the major terms enriched under BP in viruliferous *M. persicae*; “phosphotransferase activity”, “alcohol group as acceptor”, “kinase activity”, “transferase activity”, “transferring phosphorus-containing groups”, and “transferase activity” were the major terms enriched under MF; and “integral component of membrane”, “intrinsic component of membrane”, and “membrane” were the major terms enriched under CC (Figure 2b). According to the KEGG database, annotated DEPs were mainly involved in the pathways “carotenoid biosynthesis”, “apoptosis–multiple species”, “glycosphingolipid biosynthesis–ganglio series”, and “fructose and mannose metabolism”, as well as some signal pathways, such as “HIF-1 signaling pathway”, “Hippo signaling pathway”, and “AMPK signaling pathway” (Figure 4).

### 3.4. RT-qPCR Validation

RT-qPCR was performed to validate results from transcriptomic and proteomic analysis. Ten DEGs were selected, based on functional annotation, for qPCR analysis, namely *M. persicae* tubulin alpha-2 chain-like, probable isocitrate dehydrogenase (NAD) subunit alpha, trifunctional enzyme subunit alpha, cytochrome c oxidase subunit 5B, aldose 1-epimerase-like, beta-galactosidase-like, cuticle protein 65-like, general odorant-binding protein 28a, uncharacterized LOC111032711, and cuticle protein 21-like. Among the genes tested, the expression levels determined by qPCR of 80% of the DEGs were in agreement with the transcriptomic results (Figure 5a). Ten DEPs were also selected to validate the TMT-sequencing data, namely phosphatidylinositol 4-kinase beta, programmed cell death protein 2, zinc finger matrin-type protein 2, protein EFR3-like, serine/threonine-protein kinase SMG1, uncharacterized protein LOC111034038, baculoviral IAP repeat-containing protein 5-like, xylulose kinase, transmembrane protein 65, and otoferlin-like. These results also indicated that the expression levels of 80% of the DEPs were consistent between the two methods (Figure 5b); moreover, the log_2_ ratio of selected DEGs in RNA-Seq data and the ratio of selected DEPs in TMT analysis were presented in Appendix A. Thus, the results from the two omics sequencing strategies were reliable, as suggested by earlier studies [25,28,29].

## 4. Discussion

Transcriptomics and proteomics provide us with new insights for assigning a putative function to a gene by bioinformatic annotation from the perspective of mRNA and protein levels. In this study, two approaches, RNA-Seq and TMT-based quantitative proteomic analysis, were used to assess the differences between BrYV-carrying and BrYV-free aphids. Interestingly, changes in the levels of gene expression were much higher than those at the level of proteins, with DEGs and DEPs being mostly upregulated in viruliferous aphids. This difference in abundance may be related to the stability of the protein, so that only some changes in gene transcription were reflected at the protein level [30].

In previous studies, the majority of evidence indicated that poleroviruses were unable to replicate in aphids, with their virions being packaged into vesicles to complete the whole process of circulation. For successful transmission by their insect vectors, poleroviruses have to overcome midgut and accessory salivary gland barriers by interacting with receptors on the cell membrane. Based on the proteomic results, we found that there were only 18 DEPs, which may be related to the specific circulative manner or because some DEPs accumulated differently among various aphid tissues, although the overall level was balanced. Consequently, the difference between the viruliferous and nonviruliferous aphids was not significant. Directional correlation is usually performed to evaluate the relationships between levels of mRNAs and proteins. In general, there is no direct correlation between proteomic and transcriptomic results. One of the biological reasons for a low correlation may be due to posttranscriptional regulatory mechanisms [30,31,32,33].

In the current study, seven DEGs and DEPs were identified using the two omics strategies, but only xylulose kinase was upregulated at both of the mRNA and protein levels (Table 4). Although the correlation between DEGs and DEPs in transcriptomics and proteomics, respectively, was low, the enrichment results from GO and KEGG databases indicated that they were both involved in a range of metabolic processes, such as “carbohydrate metabolic process” and “single-organism carbohydrate metabolic process” for DEGs, as well as “phosphate-containing compound metabolic process”, “phosphorus metabolic process”, and “lipid metabolic process” for DEPs. Similarly, they were both associated with the “phosphorylation pathway”, which provided us with clues by which to analyze and identify the key factors in BrYV-carrying aphids.

Insect cuticle proteins are mainly divided into 12 families, based on the proteins of insects that have a complete genome sequence, with over 1% being cuticle proteins, which are essential to the composition of insect structures [34]. Previous studies have shown that insect cuticle proteins could enhance the ability of insects to adapt to harsh conditions, such as environmental stresses, insecticides, or drought conditions [35,36]. In our RNA-Seq data, at least 25 cuticle protein-like genes were obviously upregulated in viruliferous aphids, relative to nonviruliferous aphids (Table 5). Similarly, the upregulated phenomenon of insect cuticle gene expression has also been reported in BYDV-GPV-infected *R. padi* and CMV-infected *M. persicae*, as well as TYLCV- or PaLCuCNV-infected *B. tabaci*. These results indicated that the insect immune responses may be activated by these viruses and that upregulation of cuticle proteins would enhance their own defense in response to virus acquisition [22,24,25]. In addition, some studies have shown that cuticle proteins were able to be involved in the transmission of nonpersistent, semipersistent or circulative–propagative viruses [37,38,39]. For example, MpCP4 could interact with CMV-CP directly; the knockdown of CP4 transcription by RNA interference (RNAi) resulted in a decrease in virus acquisition [38]. The P2 protein of cauliflower mosaic virus (CaMV) was observed via electron microscopy to be associated with the receptor at the extreme tip of the aphid maxillary stylets, which are deeply embedded therein, similar to the cuticle proteins [37]. The novel cuticular protein, CPR1, of SBPH could bind the nucleocapsid protein of RSV, reduce the viral concentration in the hemolymph, and suppress the ability of SBPH to transmit RSV via RNAi of CPR1, indicating that the cuticle protein plays an important role in virus circulation and transmission [39]. However, the molecular mechanism by which the cuticle protein is involved in poleroviruses acquisition and transmission remains unclear. In our data, some cuticle proteins were upregulated in BrYV-carrying aphids; the function of these genes in BrYV acquisition and transmission could be verified using RNAi, which may provide a reference for the role of cuticle proteins in response to BrYV stress.

Microtubules are essential cytoskeletal polymers that are made of α-/β-tubulin heterodimers and which function in terms of cell shape, cell transport, cell motility, and cell division in eukaryotic organisms [40]. In addition, microtubules are also involved in autophagy and virus transmission, such as in porcine circovirus (PCV2), pseudorabies virus (PRV), and influenza A virus (IAV), which could promote virus replication by regulating the transport of microtubules to the nucleus [41,42,43,44]. Previous studies have demonstrated that host tubulin could interact with viral proteins involved in viral transport, such as a 48-kDa tubulin or tubulin-like protein of dengue virus 2 in C6/36 mosquito cells [45]. Proteomic analysis of SBPH salivary glands indicated that tubulin alpha-2 chain was upregulated in viruliferous aphids, relative to nonviruliferous SBPH. In addition, LsTUB could interact with RSV NS3 in vitro, with RNAi-mediated silencing of LsTUB having no effects on their feeding behavior, but the transmission efficiency decreased in dsLsTUB-treated viruliferous SBPH, which revealed that LsTUB may play a critical role in helping RSV overcome the midgut and salivary gland barriers [46]. Furthermore, at least 15 tubulin-like genes were upregulated in BrYV-carrying aphids vs. BrYV-free aphids, which means that these tubulin-like proteins may be expressed in response to virus infection by increasing their accumulation. However, whether tubulins could interact with BrYV to influence virus passage past the midgut and salivary gland barriers needs further study (Table 6).

A previous study demonstrated that 134 genes were shown to be expressed differentially in *M. persicae* carrying PLRV relative to nonviruliferous aphids [1]. Based on the GO annotation, metabolic processes (41%), cellular-protein processes (11%), and oxidation–reduction processes (11%) were discovered to be significantly enriched in the biological process aspect. Similarly, terms related to metabolic processes also exhibited significant enrichment in BrYV-carrying aphids. In the molecular function aspect, most of the DEGs in PLRV-carrying aphids were involved in catalytic activity (33%) or nucleic acid binding (21%). However, the majority of DEGs were enriched in oxidoreductase activity, structural constituent of cuticle, structural molecule activity, and peptidase activity terms in the molecular function aspect for aphids carrying BrYV. Based on the KEGG enrichment analysis in PLRV-carrying aphids, the upregulated DEGs were mainly involved in cuticle formation and development pathways. According to our RNA-Seq data, at least 25 cuticle protein-like genes were significantly upregulated in viruliferous aphids, indicating that cuticle proteins may have a potential role in mediating polerovirus transmission. With respect to the downregulated DEGs, histones and histone-modifying proteins were primarily enriched in PLRV-carrying aphids, although these pathways were not significantly enriched in our RNA-Seq data. Up until this study, the functions of cuticle proteins in polerovirus transmission have not been reported, so our data will provide a reference for studying the role of cuticle proteins in response to polerovirus stress. Interestingly, more DEGs were identified in BrYV-carrying aphids, in the comparisons with PLRV-carrying aphids. Furthermore, the majority of the GO and KEGG enrichment analyses in aphids differed in response to various polerovirus stresses, so the molecular mechanism for this phenomenon warrants further investigation.

RNAi has been considered to be a potential strategy by which to study individual gene functions in various organisms. It is a novel and safe tool widely applied in pest management. However, successful RNAi seems to be quite variable among different insect species [47], so this method is suitable for the functional validation of selected genes but is not a good choice to achieve a large-scale analysis of gene functions. In our study, differentially expressed genes or proteins could be identified using omics, based on enrichment by the GO and KEGG databases, and critical genes could be selected to explore the most suitable system for RNAi, which may be helpful in identifying functional genes involved in virus circulation.

BrYV is restricted to the phloem cells of hosts and transmitted by aphids in a circulative manner. The virions have to penetrate gut and salivary gland barriers to finally transmit to new hosts. Meanwhile, aphid saliva plays important roles in aphid–host interactions. Several salivary effectors of *M.persicae* were identified to influence aphid performance by modulating plant defense pathways, such as MpC002, Mp55, and Mp10 [48,49,50]. Moreover, 76 salivary proteins were identified from the watery saliva of *S. graminum* using transcriptome and proteome analyses by dissecting salivary glands; among them, Sg2204 was identified to suppresses wheat defense and have positive impacts on the behavior of aphids [51]. Sm9723, a candidate salivary effector which was identified in *Sitobion miscanthi*, could inhibit *Nicotiana benthamiana* defense responses and was beneficial to the fecundity, survival, and feeding behaviors of *S. miscanthi* [52]. At present, our RNA-Seq data and TMT analysis are based on the whole aphid. In the future, it will be possible to isolate the gut and stylet of aphids to identify more significant proteins.

## 5. Conclusions

In this study, two approaches were performed to identify the genes and proteins differentially expressed between viruliferous and nonviruliferous *M. persicae*. Our data indicated that either the number or fold change of upregulated DEGs and DEPs in viruliferous aphids were significantly higher than those in downregulated DEGs and DEPs, with both omics being enriched in the biological processes and pathways associated with metabolic process and phosphorylation. In addition, some cuticle proteins and tubulins were significantly upregulated in BrYV-carrying aphids, indicating that these two classes of proteins may play a crucial role in virus acquisition and transmission. Taken together, our study reveals elements of the regulation between BrYV and green peach aphids at the mRNA and protein levels, which will be beneficial for screening key factors in aphids in response to BrYV stress and provides a crucial clue to exploring the molecular mechanism of the transmission of BrYV by *M. persicae*.

## Figures and Tables

**Figure 1 biology-12-00908-f001:**
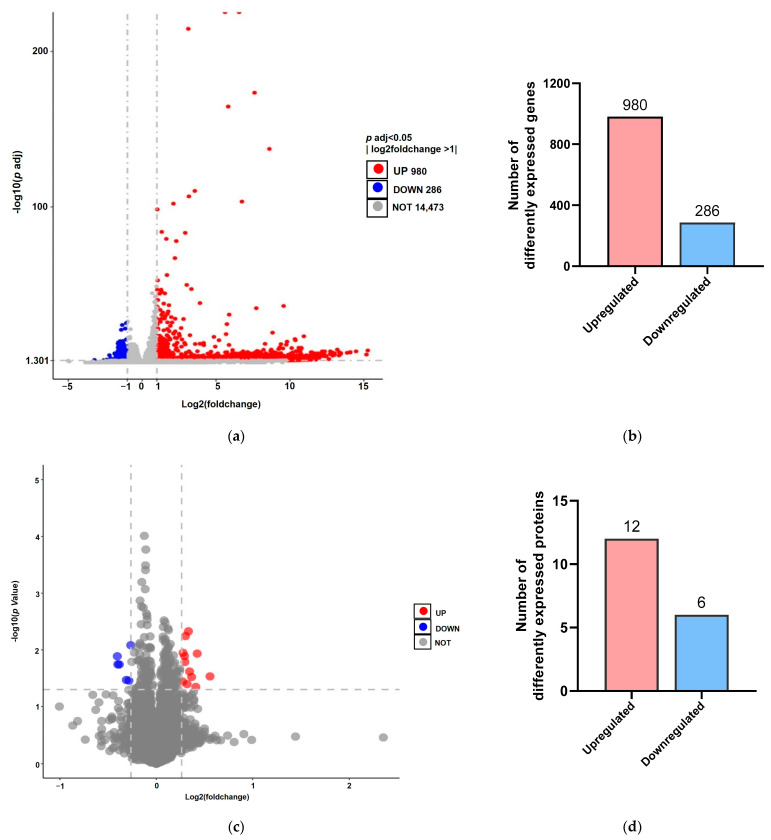
Genes and proteins differentially expressed between viruliferous and nonviruliferous aphids. Volcano plot presented the distribution of DEGs (**a**) and DEPs (**c**) in viruliferous aphids vs. nonviruliferous aphids, averaged over three biological replicates. Red points represent significantly upregulated genes or proteins, blue points represent significantly downregulated genes or proteins, whereas gray points represent nondifferentially expressed genes or proteins. DEGs were identified using |log_2_ (fold change, FC)| > 1 and *P p* adj < 0.05 as the criteria, and DEPs were identified using FC > 1.2 (upregulated > 1.2 or downregulated < 0.83) and *p* value < 0.05 as the criteria. (**b**) The number of DEGs in viruliferous aphids relative to nonviruliferous aphids; (**d**) The number of DEPs in viruliferous aphids relative to nonviruliferous aphids.

**Figure 2 biology-12-00908-f002:**
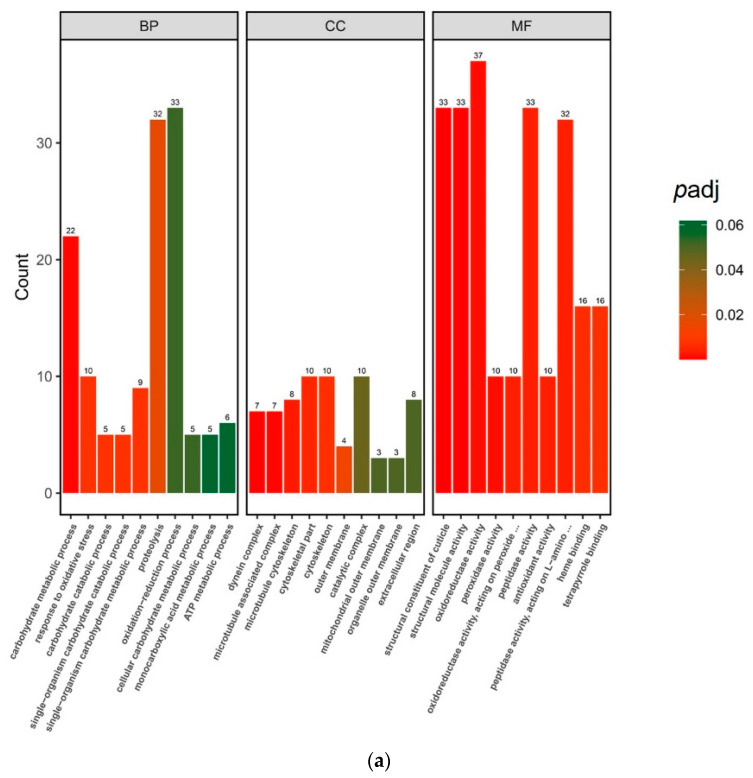
GO enrichment analysis of DEGs (**a**) and DEPs (**b**) in viruliferous aphids. The X-axis represents the enriched GO terms (*p* < 0.05) identified in the three categories’ biological process (BP), cellular component (CC), and molecular function (MF). The Y-axis represents the number of genes and proteins.

**Figure 3 biology-12-00908-f003:**
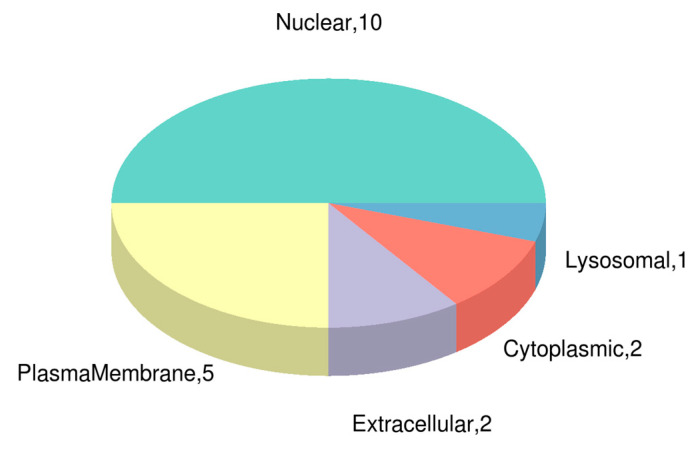
Subcellular localization of DEPs in viruliferous aphids. The number represents the number of DEPs which are located in nuclear, plasma membrane, extracellular, cytoplasmic, or lysosomal regions. Specific differentially expressed protein information is shown in Appendix A.

**Figure 4 biology-12-00908-f004:**
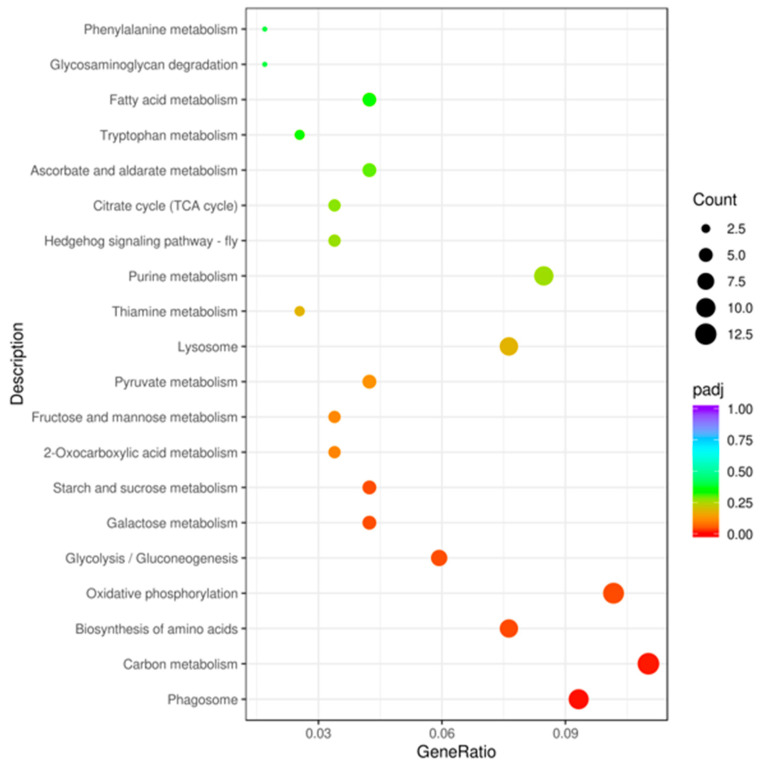
KEGG enrichment analysis of DEPs in viruliferous aphids. The X-axis represents the ratio of the number of DEPs annotated in this pathway term to the number of all DEPs annotated. The bubble color indicates the significance of KEGG pathways. Only functional categories and pathways with *p*-values under the threshold of 0.05 were considered to be significant.

**Figure 5 biology-12-00908-f005:**
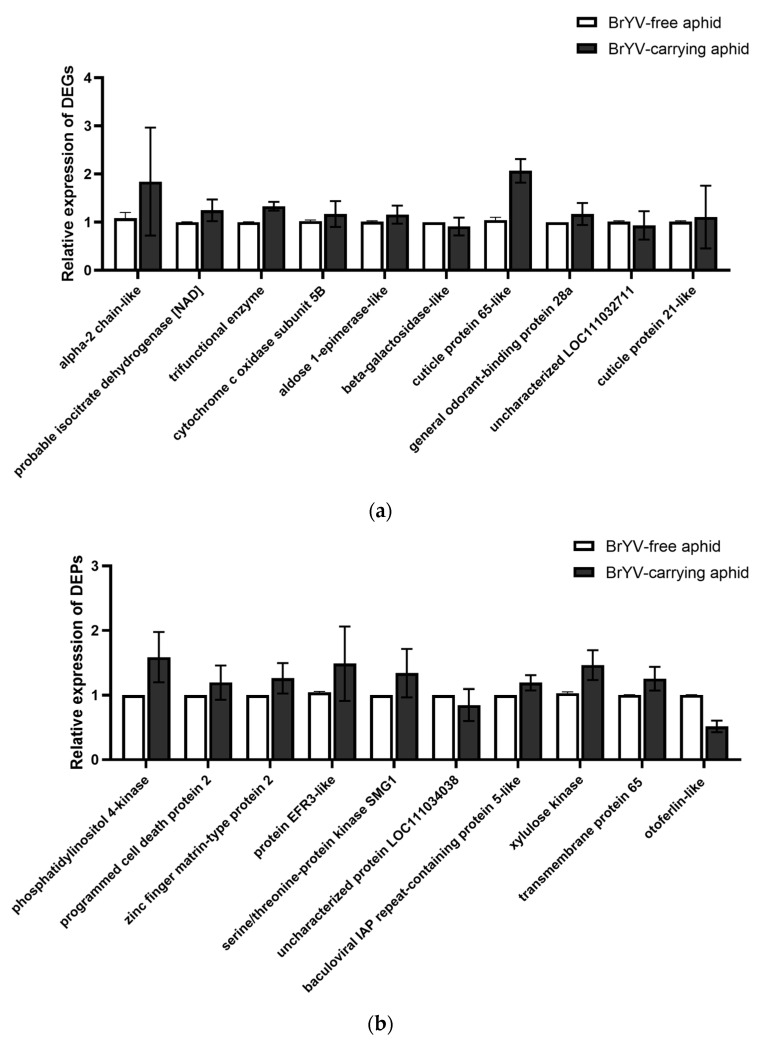
qPCR validation of selected genes (**a**) and proteins (**b**) differentially regulated in viruliferous aphids. The error bar indicated the relative expression difference of three biological replicates between viruliferous and nonviruliferous aphids.

**Table 1 biology-12-00908-t001:** Transcriptomic parameters of *Myzus persicae*.

Items	Viruliferous Aphids	Nonviruliferous Aphids
Raw reads number	53,943,971	56,380,878
Clean reads number	53,190,385	55,432,735
Q30 of clean reads (%)	92.82	93.36
Unique mapped reads (%)	48,830,235 (91.79)	51,066,134 (92.12)

**Table 2 biology-12-00908-t002:** Proteomic statistics of *Myzus persicae*.

Items	Number
Total spectrum	725,126
Mapped spectrum	71,746
Unique peptides	34,831
Identified proteins	5567

**Table 3 biology-12-00908-t003:** KEGG enrichment analysis of DEGs in viruliferous aphids.

Description ^a^	Number of Genes		Gene Ratio ^b^	*p* Value	Pathway ID
	Up	Down			
Phagosome	9	2	11/118	5.36 × 10^−5^	api04145
Carbon metabolism	12	1	13/118	0.000164174	api01200
Biosynthesis of amino acids	8	1	9/118	0.001231352	api01230
Oxidative phosphorylation	12	0	12/118	0.001716266	api00190
Glycolysis/Gluconeogenesis	7	0	7/118	0.002545986	api00010
Galactose metabolism	5	0	5/118	0.003196235	api00052
Starch and sucrose metabolism	3	2	5/118	0.003196235	api00500

^a^ Pathways enriched according to the KEGG database. ^b^ The ratio of DEGs annotated in this pathway term relative to the number of all DEGs.

**Table 4 biology-12-00908-t004:** Correlation between mRNA and protein levels for seven DEGs in BrYV-carrying aphids.

NCBI Reference Sequence	Log_2_ (Transcript Ratio)	Protein Ratio	Annotation
XM_022317504.1	−0.70	1.29	bifunctional lycopene cyclase/phytoene synthase-like
XM_022319060.1	−0.54	1.26	acetyl-coenzyme A transporter 1-like
XM_022319915.1	−0.64	1.22	baculoviral IAP repeat-containing protein 5-like
XM_022324126.1	0.41	1.21	xylulose kinase
XM_022310409.1	0.68	0.83	uncharacterized family 31glucosidase KIAA1161-like
XM_022324319.1	0.31	0.77	6-phosphofructo-2-kinase/fructose-2,6-bisphosphatase-like
XM_022306530.1	0.62	0.76	opsin, ultraviolet sensitive-like

**Table 5 biology-12-00908-t005:** Upregulated expression of cuticle protein-like genes in viruliferous aphids.

Gene_ID	Gene_Description	Log_2_ Fold Change
111035272	cuticle protein 65-like	6.56
111039232	cuticle protein 7-like	7.60
111042286	cuticle protein 12.5-like	5.82
111034582	cuticle protein 19-like	9.57
111031114	Endocuticle structural glycoprotein SgAbd-8-like	1.57
111039055	cuticle protein 21-like	2.64
111034580	cuticle protein 7-like	2.24
111039059	cuticle protein 7-like	2.22
111039229	larval cuticle protein A3A	2.28
111039056	cuticle protein 7-like	1.54
111034589	cuticle protein 7-like	1.55
111034588	cuticle protein 7-like	1.53
111032404	cuticle protein 38-like	1.69
111034590	cuticle protein-like	1.51
111042794	cuticle protein 12.5-like	1.68
111039058	cuticle protein 7-like	2.97
111037705	cuticle protein 19-like	1.30
111039066	PF00379: Insect cuticle protein	1.97
111034581	cuticle protein 7-like	2.40
111037704	cuticle protein-like	1.03
111039052	cuticle protein 19-like	1.02
111031121	endocuticle structural glycoprotein SgAbd-4-like	1.38
111039057	cuticle protein 7-like	1.70
111035301	cuticle protein 19.8-like	1.51
111034586	cuticle protein-like	1.28

**Table 6 biology-12-00908-t006:** Upregulated expression of tubulin-like genes in viruliferous aphids.

Gene_ID	Gene_Description	log_2_ Fold Change
111038858	tubulin beta chain-like	9.23
111037228	tubulin beta chain-like	8.21
111040080	tubulin glycylase 3A-like	6.11
111038861	tubulin alpha-4 chain-like	15.26
111033819	tubulin glycylase 3A-like	11.42
111035160	tubulin glycylase 3A-like	7.49
111038612	tubulin beta chain-like	8.73
111035562	tubulin beta chain-like	11.70
111033210	tubulin glycylase 3A-like	5.90
111038008	tubulin glycylase 3A-like	11.34
111028369	gamma-tubulin complex component 3 homolog	5.02
111042666	tubulin alpha-2 chain-like	4.76
111038281	tubulin glycylase 3A-like	8.55
111034561	tubulin polyglutamylase TTLL6-like	8.19
111041754	probable tubulin polyglutamylase TTLL1	2.64

## Data Availability

The data presented in this study are available within the article.

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
