# Peer review of "Transcriptomic and Proteomic Analyses of Myzus persicae Carrying Brassica Yellows Virus"

_biology, 2023, doi:10.3390/biology12070908_

Round 1

Reviewer 1 Report

This is an interesting study which provides valuable information on the aphid-mediated transmission of BrYV. 

I have a number of suggestions which authors should carefully consider to furthe improve their manuscript. 

* Which criteria authors used to select the candidate genes for validation of RNA-seq data ? Please elaborate it in the methods section.

* Did authors map the RNA-seq reads against viral genome ? If not, please do so to further validate your findings.

* Please include the qPCR results of viral expression among viruliferous aphids.

* Figure 5 (a,b): The expression of various genes/proteins should be drawn against that obtained by omics analysis to make a more clear comparison about concordant expressions.

* Supplementary table S1, Column 2: The column title should be “Gene name” rather than “Test name”.

* Supplementary table S1: Please provide the actin gene primers sequences used for qPCR experiment.

* Please provide a supplementary table associated with individual Q20 and Q30 values for six samples.

* Please carefully revise the manuscript to omit typos and grammatical mistakes.

* Please carefully revise the manuscript to omit typos and grammatical mistakes.

Reviewer 2 Report

Please consider revising some grammatical issues (some examples lines 12,14,26,41)

Please also ensure acronyms are defined such as SDT, line 185

Is it possible to isolate particular organs of aphids? ie gut, or stylet? Perhaps more significant proteins would be identified in these parts specifically? If it is possible, should this be discussed?

There are a few grammatical issues that I think should be addressed

(some examples lines 12,14,26,41)

Line 26: is M. persicae altering its proteome and transcriptome? or are the "omes" of this insect being altered during viral infection
